# Psychometric Properties of the 11-Item De Jong Gierveld Loneliness Scale in a Representative Sample of Mexican Older Adults

**DOI:** 10.3390/healthcare11040489

**Published:** 2023-02-08

**Authors:** Liliana Giraldo-Rodríguez, Teresa Álvarez-Cisneros, Marcela Agudelo-Botero

**Affiliations:** 1Instituto Nacional de Geriatría, Mexico City 10200, Mexico; 2Centro de Investigación en Políticas, Población y Salud, Facultad de Medicina, Universidad Nacional Autónoma de México, Mexico City 04510, Mexico

**Keywords:** loneliness, psychometric properties, scale, older adults, Mexico

## Abstract

Recent studies have focused on the negative effects of loneliness on health and quality of life in older adults. The De Jong Gierveld Loneliness Scale (DJGLS) has been widely used and has proven to be a valid and reliable instrument for loneliness assessment. However, research on this topic and on the validation of measurement scales among the older population is still incipient. The objective of this study was to examine the psychometric properties of the Spanish version of the 11-item DJGLS in Mexican older adults. Data from a representative sample of cognitively intact older adults aged 60 years and over (mean, standard deviation [SD]) age = 72.0 years (SD 8.1) from two Mexican cities (*n* = 1913), interviewed face to face at their homes during 2018–2019, were analyzed. The psychometric properties of the DJGLS were examined, including (1) construct validity, examined by Exploratory Factor Analysis (EFA) and Confirmatory Factor Analysis (CFA), discriminate validity and convergent validity, (2) reliability, calculated using Cronbach’s alpha. The overall data quality was high, and the scaling assumptions were generally met with few exceptions. Using EFA and CFA, the findings showed that the DJGLS presents a two-factor structure (Social Loneliness and Emotional Loneliness), with 11 items that explain 67.2% of the total variance. Reliability is adequate at the full-scale level (Cronbach´s α = 0.899), also for the two subscales Social and Emotional Loneliness (Cronbach´s α = 0.892 and 0.776, respectively). These results highlight that most participants with a low score for depressive symptoms and or with a high social support score belonged to the “No loneliness” group. The results showed that the Spanish version of the 11-item DJGLS is adequate for use in Mexican older adults and should be used not only for loneliness screening but also for social and emotional loneliness assessment.

## 1. Introduction

Loneliness in older adults has been considered by some authors as the pandemic of the 21st century because of its negative effects on health and quality of life, comparable to those of well-established risk factors such as alcohol consumption, smoking, obesity and frailty [1]. Loneliness is defined as experiencing the sensation of being alone. However, this poses a cognitive discrepancy between the social relationships that an individual possesses and their own preference. An individual may experience this feeling despite being surrounded by others because the physical presence of other people cannot suffice, revealing that the sensation of having significant connections is necessary to not feel alone [2].

Older adults may experience loneliness as a result of life experiences, such as the loss of a partner, friends or colleagues, the distancing from significant others (for example, the departure of daughters or sons from home or retirement), but also as a result of the loss of functional capacity [3]. Some of the negative effects of loneliness on older people`s health are a functional decline and an increased mortality risk [4,5], as well as a higher likelihood of depression and other mental conditions [6,7] and decreased quality of life [8], life satisfaction, sleep and cognition [9].

In view of these effects, the accurate measurement of loneliness using scales that allow consistent and comparable results for its identification in clinical and epidemiological settings is crucial. Numerous scales have been developed to measure loneliness, varying in dimensions and number of items [10] and using direct and indirect measures [11]. Direct measures are those that inquire directly and usually through a single question about loneliness. Indirect measures cover multiple aspects of the concept of loneliness; some examples are the UCLA Loneliness Scale (Version 3) [12] and the De Jong Gierveld Loneliness Scale (DJGLS) [13]. The DJGLS is one of the most widely used since 1987 [11,14,15,16] and was developed to assess severe and less severe feelings of loneliness [17].

The DJGLS multidimensional model of loneliness proposes that loneliness is not a state, but a process in which both the characteristics of the individuals (such as social skills) and the characteristics of their social network intertwine, taking into consideration cultural influences such as social norms and achievement orientation [14]. The DJGLS is based on the mediating cognitive processes between the characteristics of the social network and the experience of loneliness, emphasizing the importance of personal perceptions and the interpretation of social relationships, which do not necessarily depend on the real number of social relationships [13]. This scale consists of five positive and six negative items. The positive items assess feelings of belongingness and refer primarily to the absence of a broader category of acquaintances, colleagues and friends—social loneliness—whereas the negatively formulated items refer primarily to the fact that a partner, a special or best friend is sorely missed, missing relationships—emotional loneliness [18].

While the predictive validity of the DJGLS has been widely documented, showing high internal consistency and reliability and suitability for use with different age groups and populations [19], the discussion regarding its factorial structure is still ongoing. Some authors consider the internal structure of the scale controversial, arguing that it has a univariate structure [20,21], while other authors disagree, highlighting that the scale comprises two factors: one looking into Social Loneliness and the other looking into Emotional Loneliness [19,22,23]. However, this division into two factors could be a result of a method’s bias associated with the negatively worded items [17,20,24]. Because of this, analyzing the construct validity (discriminant and convergent validity) of each of the two possible structures and of the scale as a whole was cardinal for our analyses. This type of analysis help assures that the scale and each sub scale measure loneliness as a whole, but also social and emotional loneliness individually.

Several studies have been conducted to determine the feasibility of translating the DJGLS into other languages and populations. The scale’s reliability and validity in the older population have been assessed in China [22,25], Spain [17,26], Iran [23,27], Malaysia [28], Israel [29], the Netherlands [30] and Canada [24]. In Latin America, these validations were carried out only in Peru [31] and Chile [32].

In Mexico, despite the rapid aging of the population, no studies have evaluated the psychometric properties of the DJGLS in older adults. In 2020, people aged 60 and over represented 12% of the total population and are expected to reach 23% by 2050 [33]. In the country, the prevalence of loneliness in adults aged ≥65 years using a single item measure has been estimated to be 32.4%. Findings have also suggested an association between loneliness and mortality [5]. Loneliness is an emerging problem in the Mexican older adult population and is expected to rise as the number of people in this age group increases. Given the high prevalence of loneliness, its negative effects and the lack of validated scales in this country allowing an accurate detection of loneliness in the older Mexican population and cultural comparisons, the objective of this study was to examine the psychometric properties of the Spanish version of the 11-item DJGLS in Mexican older adults living in the community.

## 2. Materials and Methods

### 2.1. Data Collection

The data for the study were collected as part of a larger project: Health and Living Conditions of Older Adults conducted in Mexico City and Xalapa, Veracruz. Both cities are important metropolitan areas, belonging to two of the federal entities in the country with the largest proportion of adults who were 60 years old and over in 2020. Mexico City has 16.2% and Veracruz 14.4% of adults aged 60 years and over [34]. The study is based on a probabilistic sampling design representative of the population aged 60 years and older in both cities. In total, 2341 homes were visited; once an older person in each home was identified, inclusion criteria to participate in the project included being 60 years and over, not having cognitive impairment and being regular residents of the selected household. Structured, face-to-face, pen-and-paper interviews were carried out at the respondents’ homes. Quality assurance procedures were implemented during fieldwork. The Spanish version of the Mini-Mental Status Examination (MMSE) was completed in order to find whether the participants showed signs of cognitive impairment. Those with a score <24 were excluded from the study. Trained interviewers with previous experience in interviewing older adults were chosen to collect the data during the months from September 2018 to January 2019. Interviews were 25 to 40 minutes long. A total of 1923 direct in-person interviews were conducted (response rate = 82.1%). For the current analyses, the responses of 10 participants were excluded, as these subjects did not complete the interview. Thus, the final analytical sample comprised 1913 participants.

### 2.2. Measurements

A comprehensive questionnaire was purposely developed to investigate the demographic, socioeconomic and health characteristics of the study population, also considering the social support networks and intra-family relationships.

#### 2.2.1. Loneliness

Loneliness was assessed by the DJGLS which is based on the cognitive theoretical model of loneliness of de Jong Gierveld [13]. The original scale has eleven items: five positive (1, 4, 7, 8, 11) and six negatives (2, 3, 5, 6, 9, 10), with three answer options: “yes”, “more or less” and “no”. For the purpose of this study, the responses were operationalized as follows: three-category responses were transformed into dichotomous responses. Responses indicating a (certain) feeling of loneliness were assigned a score of 1 loneliness point. That is, if the response “more or less” or “yes” was given to a negatively formulated item or if the response “no” or “more or less” was given to a positively formulated item, a scale point was assigned. According to this procedure, the “more or less” answers were not viewed as neutral answers, but as indicators of loneliness. The other answers were assigned a zero score. Items from the DJGLS were converted into questions, including the following sentence at the beginning of each: “The following questions refer to how you feel about different aspects of your life” (Table 1).

#### 2.2.2. Depression

The depressive symptoms were measured by the 15-item Geriatric Depression Scale (GDS-15) [35,36], which was previously validated in the Mexican population showing reliability and validity [37]. The GDS-15 scale includes dichotomous responses (yes/no); a score equal to or greater than 5 positive items indicates depressive symptoms.

#### 2.2.3. Social Support

Social support (marital, filial and family support) was measured using the Spanish version of the Ekas and collaborators’ scale [38]. This scale combines three different sub-scales of support: from a spouse (marital support), from children (filial support) and from the family (family support). Every sub-scale includes four questions with information about how individuals feel, whether they trust anyone, if they feel that they are heard when they have a problem and if they feel disappointed [39].

### 2.3. Statistical Analyses

We examined data distribution, completeness, as well as out-of-range data. Using descriptive statistics, we were able to obtain the percent of missing data for each item of the DJGLS. Missing values were not imputed. In order to evaluate responses, to each of the items, distribution, means, standard deviations (SD), skewness and kurtosis were calculated. Several statistical methods were applied to test the psychometric properties of the scale. These are presented in the following section.

#### 2.3.1. Validity

##### Construct Validity

Construct validity was examined using exploratory and confirmatory factor analysis, discriminate validity and convergent validity. The total sample was randomly divided into two subsamples for the factor analysis.

Exploratory factor analysis

The item responses were dichotomized, and accordingly, Exploratory Factor Analysis (EFA) was performed over the tetrachoric correlation matrix of the first subsample [40,41,42], using the Principal Component Analysis (PCA) method and oblique (Robust Promin) rotation [43,44]. For assessing the number of dimensions of the factorial structure, the optimal implementation of the Parallel Analysis (PA) was used [45]. The sampling adequacy of the correlation matrix was evaluated with the Bartlett’s test of sphericity and the Kaiser–Meyer–Olkin (KMO) statistic, using the characterization criterion of Kaiser [46], which considers a value from 0.70 to 0.79 “acceptable”, from 0.80 to 0.89 “good and greater”, and of 0.90 or more “excellent”. Component loads greater than 0.30 were stablished in order to consider the item as part of the factor.

Confirmative factor analysis

Subsequently, a Confirmatory Factor Analysis (CFA) was performed on the second subsample using the weighted least squares with mean and variance (WLSMV) method. This suits better non-normal categorical or dichotomous variables [47]. Using CFA, we evaluated two different models: a unidimensional model (where all the items contribute to a single factor) and a two-dimensional model (Social and Emotional Loneliness) in order to select the better suited for the studied sample. Evaluation of the goodness of fit of the models was carried out considering the cut-off points suggested by Hu and Bentler [48]: χ^2^ significant, Tucker–Lewis Index (TLI) >0.90, Comparative Fit Index (CFI) >0.95, Root-Mean-Square Error of Approximation (RMSEA) <0.06. Since the model estimation was carried out by the WSLMV method, the comparison of the models was performed from the goodness of fit indices, selecting the one that showed the best fit for the data.

Discriminant and convergent validity

The discriminant validity of the DJGLS was tested by estimating the strength of the correlation between the two factors (Social Loneliness and Emotional Loneliness). The Composite Reliability (CR) and Average Variance Extracted (AVE) statistics were evaluated. Cutoff points for CR > 0.70 and AVE > 0.70 were used, according to Fornell and Larcker [49]. Our hypothesis postulated that the factors were independent of each other; therefore, a modest to moderate correlation between the two factors was expected.

Criterion-based convergent validity was assessed using known-groups validity. This design allowed us to test whether this psychometric measure correctly discriminates between groups with dissimilar unrelated variables [50,51]. We used a two-sample *t*-test looking for differences in means with unequal variances, comparing the means of two related indicators between the “no loneliness” and the “high/very high loneliness” groups. According to the DJGLS manual, the “no loneliness” group was defined as having a DJGLS score <3, while the “high/very high loneliness” group was defined as having a DJGLS score >8 [52]. The indicators related to loneliness tested for were depressive symptoms and social support.

#### 2.3.2. Reliability

##### Internal Consistency

The Cronbach’s alpha coefficient was applied to assess the internal consistency of the scale and each resulting dimension [53,54]. Alpha values equal to 0.70 or higher were considered acceptable [55,56]. 

### 2.4. Software Used for Analysis

EFA was performed using Factor software, version 11.05.01 [57]; CFA was performed using Mplus, version 7.4 [58], and the two-sample *t*-test with unequal variances were performed using Stata version 16.1 [59].

## 3. Results

Our working sample consisted of 1913 Mexican older adults, of which 64% were women, with a mean age of 72.0 years (SD 8.1) and general low educational attainment, with 51% of the sample having completed primary-level education or without schooling (Table 2).

### 3.1. Item Descriptive Statistics

Table 3 presents the descriptive statistics of each item of the DJGLS. The item’s mean (which, being the items dichotomous, can be interpreted as the proportion of people who responded positively) ranged from 0.153 (for item 4) to 0.409 (for item 6), while the standard deviation ranged from 0.347 (for item 10) to 0.492 (for item 6). Items 1, 3, 4, 9, 10 and 11 presented positive asymmetry (greater than one), and all items showed a leptokurtic distribution. The completion rate was high (99.1–99.9%), and the items with the highest missing values were item 3 (0.9%), item 2 (0.8%) and item 7 (0.6%).

### 3.2. Construct Validity

#### 3.2.1. Exploratory Factor Analysis

The results of the EFA are shown in Table 4. Statistical analyses of the data matrix fit confirmed that the data adequately fit the model (Bartlett’s statistics = 6633.5, degrees of freedom (df) = 55, *p* < 0.001 and KMO = 0.849).

The analysis yielded a structure of two factors that had an eigenvalue greater than 1 (Factor 1 = 5.69 and Factor 2 = 1.70), which can be seen in Appendix A, and which, together, explained 67.2% of the variance of the complete model (Factor 1 = 51.7%, Factor 2 = 15.5% of the total explained variance). The initial communalities of the items were between 0.461 and 0.780. After performing the promin robust rotation, the items corresponding to Factor 1 “Emotional Loneliness” (items 2, 3, 5, 9 and 10) had factor loadings between 0.724 and 1.001. The items that made up Factor 2 “Social Loneliness” (items 1, 4, 7, 8 and 11) presented factorial loads between 0.521 t and 0.971. Item 6 behaved in a non-specific way, contributing, for both factors, with low factor loads (0.419 for “Social Loneliness” and 0.389 for “Emotional Loneliness”).

#### 3.2.2. Confirmatory Factor Analysis

Using the second subsample, the CFA was carried out considering the factorial structure proposed by the EFA. The data showed a good fit for the model according to the goodness-of-fit indices, except for the χ^2^ index (χ^2^ = 183.981, degrees of freedom (df) = 42, *p* > 0.001; CFI = 0.966; TLI = 0.955; RMSEA = 0.059, confidence interval (CI) 95% [0.051–0.068]). The CFA showed that the factor loadings of the items assigned to the “Emotional Loneliness” factor (items 2, 3, 5, 6, 9 and 10) had coefficients between 0.424 and 1.236 (item 2 was used as a reference, and its coefficient was set to 1) (Figure 1). Items conforming the “Social Loneliness” factor (items 1, 4, 6, 7, 8 and 11) contributed with coefficients between 0.516 and 1.090 (item 1 was used as a reference, and its coefficient was set to 1). All coefficients were high and statistically significant. Again, item 6 showed the lowest factor loading for both factors (0.424 Emotional Loneliness and 0.516 Social Loneliness).

As the EFA showed that item 6 could be related to both factors, three forms of the test were compared: the one-dimensional model, the original two-factor model, described by De Jong Gierveld and Van Tilburg [52] and the two-factor model, with item 6 contributing to both factors. The WLSMV estimation method does not allow a statistical comparison between models through the information indices, so only visual comparisons of the global adjustment indices were carried out (Table 5). The results highlighted that the two-factor model presented a better fit to the data compared to the one-dimensional model.

#### 3.2.3. Discriminant Validity

The results of discriminant validity are shown in Table 6. The correlation between the two factors was low (r = 0.319). In addition, the CR and AVE indices showed a value above the recommended cut-off point, concluding that both factors are statistically different and measure different constructs.

#### 3.2.4. Convergent Validity

To analyze the convergent validity of the scale, we used the depressive symptoms score and the three subscales of social support. The results shown in Table 7 suggest that, as expected, the depressive symptoms score was low in the “no loneliness” group (mean = 1.81) when compared to the “high/very high loneliness” group (mean = 4.55). These differences were statistically significant (t = −21.5, df = 1384.18, sig < 0.001). When applying the scale to the three subscales of social support (marital, filial and familiar support), the results showed that social support was higher in the “no loneliness” group when compared to the “high/very high loneliness” group. All these differences were also statistically significant for each of the subscales (marital support t = 5.423, df = 617.078, *p* < 0.001; filial support t = 10.941, df = 1337.95, *p* < 0.001; familiar support t = 8.231, df = 863.032, *p* < 0.001).

### 3.3. Internal Consistency

As shown in Table 3, the reliability of the total scale was good (Cronbach’s α = 0.899). Similarly, the reliability of the “Social Loneliness” subscale (Cronbach’s α = 0.892) was good, and that of the “Emotional Loneliness” subscale (Cronbach’s α = 0.776) was sufficient.

## 4. Discussion

The aim of this study was to address the psychometric properties of the DJGLS scale for loneliness in a sample of older adults from Mexico. This scale was previously translated to Spanish [26] and validated in other Spanish-speaking countries [17,26,31,32]. The results showed that this version of the scale has adequate psychometric characteristics for its use, and its reliability was found to be adequate (Cronbach’s α = 0.899). The findings from these analyses also showed that the DJGLS has a two-factor structure, comprising two subscales, Social Loneliness and Emotional Loneliness, and both components have adequate reliability, while the Cronbach’s alpha of the Social Loneliness subscale is α = 0.892, and the Cronbach’s alpha of the Emotional Loneliness subscale is α = 0.776. Therefore, this scale can be used to measure overall loneliness as well as emotional and social loneliness. The DJGLS also showed convergent validity with measures of depressive symptoms and social support, as has been reported previously in the literature [26,31], suggesting that the construct measured by this test is appropriate to be used as an indicator of loneliness.

This is not the first attempt to adapt and validate scales to measure loneliness in the older Mexican population. Acosta and collaborators [60] adapted the ESTE scale, an instrument of 34 items where 6 factors explain 58.9% of the total variance, and Montero-López [61] developed the IMSOL-AM scale consisting of 20 items grouped into two factors explaining 57.7% of the total variance. Notwithstanding, the strengths of this study include the validation of a shorter, highly reliable instrument which is widely used in other countries, allowing clinical and epidemiological comparisons between different populations and subgroups.

The factorial structure of two subscales found in the Mexican population is similar to that reported by other authors [19,22,23]. The distinction between the two types of loneliness is cardinal, as it helps to distinguish social factors, such as the number of social relationships, from emotional factors, such as the perceived quality of these relationships, and their interaction with life history, identity, personality, socioeconomic status, etc., highlighting loneliness as a complex phenomenon [62,63], with independent and temporally stable predictors and developmental pathways [64].

An additional finding of our study was that item 6, “I find my circle of friends and acquaintances too limited”, shared variance with both the Social and the Emotional Loneliness subscales. Although the reason for this particularity is not very clear, it has been argued that item 6 is ambiguous and difficult to place in either the Emotional or the Social Loneliness subscale [65]. Adaptations of the DJGLS scale from other countries have found a similar behavior of different items [17]. For example, some studies showed a low discrimination capacity of item 3 [17,22]. These differences highlight a possible influence of cultural factors. For this reason, we consider the need for future research to focus on performing a Differential Item Functioning (DIF) analysis of the scale in the Mexican population.

Given that this scale addresses loneliness from a cognitive perspective (not only from that of the quantification of an individual’s social networks) and is also a short scale (11 items), easy to perform, we support its wider use in the clinical setting and in epidemiological studies. Its use will allow the identification of loneliness in the older population. This is relevant as policy and clinical interventions for its prevention and detection of its negative outcomes can only be introduced with its accurate diagnosis and reliable statistics reflecting its burden. For example, a recent systematic review of interventions for loneliness prevention, showed that group interventions and animal-assisted therapies significantly reduced the negative cognitive effects of loneliness, improving the feelings of perceived loneliness and its related factors such as quality of life, psychological well-being, socialization and depressive symptoms [66].

Furthermore, the results from this study imply that because of its two-factor structure, interventions could be focused on decreasing Social or Emotional Loneliness. As the first looks into the number of interactions and the second into the quality or deepness of the relationships, interventions should be tailored to the specific predominant type of loneliness. Further studies in Mexico should focus on the implementation of these interventions.

### Limitations of the Study

Our study has some limitations. Despite using a representative sample of two cities in Mexico, this is not a nationally representative sample. Strictly, the results cannot be generalized to the general population; therefore, we suggest that future studies explore these findings in a broader population, focusing on how sociocultural aspects can influence the experience of loneliness in older adults. Another limitation includes the exclusion of older adults with cognitive impairment. We acknowledge that by this, we excluded a group with an increased risk for experiencing loneliness. While our findings showed that the psychometric properties of the scale are sufficient for this population, to strengthen the analysis, other validation techniques could be performed in future studies, e.g., with the Rasch model. In our study, we did not assess the possible social-desirability bias derived from face-to-face interviews. Further research comparing results obtained using other data collection techniques such as Computer-Assisted Personal Interviews (CAPI) is suggested. Lastly, it would also be relevant to assess content, concurrent and predictive validity to ensure the accuracy of the information obtained with this scale.

## 5. Conclusions

In conclusion, the Spanish version of the 11-item DJGLS shows adequate psychometric characteristics for its use in Mexican older adults. We suggest the use of this scale for screening loneliness as a whole but also its social and emotional components. As this is a relatively simple instrument to use, it is highly recommended for use in non-institutionalized older adults, allowing comparisons between countries, different populations and subgroups. Despite this study being conducted in Mexico, the validation of this scale shows that it could be used in other Latin American countries where Spanish is spoken.

Finally, this study reinforces the existing evidence of the reliability and validity of the DJGLS, so its use in different groups of older adults in Mexico is highly recommended. The broader use of this scale in Mexico would facilitate international comparisons focusing on loneliness and would help standardize a measure of its burden as well as of the impact of policies to decrease it.

## Figures and Tables

**Figure 1 healthcare-11-00489-f001:**
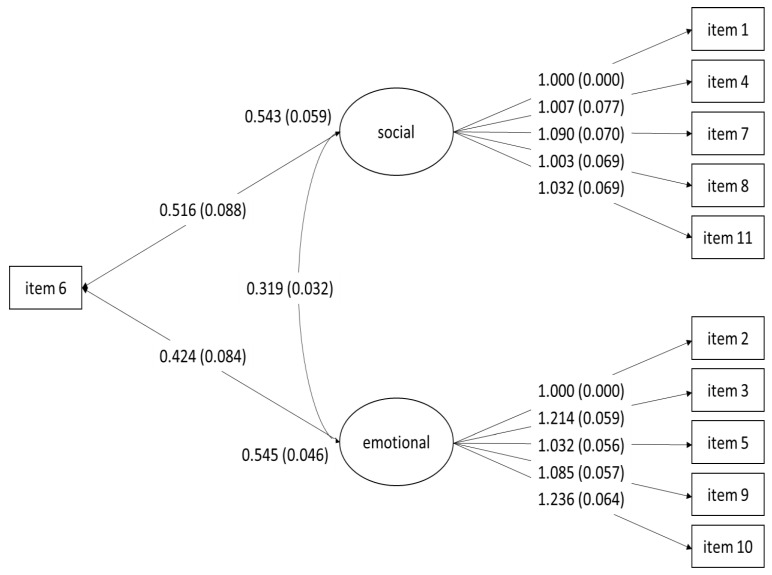
Confirmatory Factor Analysis of the two-factor model.

**Table 1 healthcare-11-00489-t001:** The 11 items of the DJGLS.

Item	Items Wording (English/Spanish)	
1	There is always someone I can talk to about my day-to-day problems. ¿Siempre hay alguien con quien puede hablar de sus problemas diarios?	+
2	I miss having a really close friend. ¿Echa de menos tener un buen amigo de verdad?	-
3	I experience a general sense of emptiness. ¿Siente una sensación de vacío a su alrededor?	-
4	There are plenty of people I can lean on when I have problems. ¿Hay suficientes personas a las que puede recurrir en caso de necesidad?	+
5	I miss the pleasure of the company of others. ¿Echa de menos la compañía de otras personas?	-
6	I find my circle of friends and acquaintances too limited. ¿Piensa que su círculo de amistades es demasiado limitado?	-
7	There are many people I can trust completely. ¿Tiene mucha gente en la que confiar completamente?	+
8	There are enough people I feel close. ¿Hay suficientes personas con las que tiene una amistad muy estrecha?	+
9	I miss having people around me. ¿Echa de menos tener gente a su alrededor?	-
10	I often feel rejected. ¿Se siente abandonado a menudo?	-
11	I can call on my friends whenever I need. ¿Puede contar con sus amigos siempre que lo necesita?	+

Notes. + = positively formulated items; − = negatively formulated items.

**Table 2 healthcare-11-00489-t002:** Sociodemographic characteristics of the sample (*n* = 1913).

Characteristics	*n*	%
Sex		
Male	689	36.0
Female	1224	64.0
Age group		
60 to 64	407	21.3
65 to 69	419	21.9
70 to 74	395	20.6
75 to 79	318	16.6
80 and over	373	19.5
Missing	1	0.1
Educational Attainment		
None	190	9.9
up to Primary	787	41.1
up to High School	543	28.4
College and above	373	19.5
Missing	20	1.1

**Table 3 healthcare-11-00489-t003:** Descriptive statistics of the DJGLS items.

	Items	Descriptive Statistics				
		Mean	SD	Skewness	Kurtosis	% Missing Data
1	There is always someone I can talk to about my day-to-day problems	0.204	0.403	1.471	3.163	0.2
2	I miss having a really close friend	0.353	0.478	0.615	1.378	0.8
3	I experience a general sense of emptiness	0.236	0.424	1.243	2.546	0.9
4	There are plenty of people I can lean on when I have problems	0.153	0.360	1.930	4.725	0.1
5	I miss the pleasure of the company of others.	0.380	0.485	0.497	1.246	0.4
6	I find my circle of friends and acquaintances too limited	0.409	0.492	0.369	1.136	0.4
7	There are many people I can trust completely	0.376	0.485	0.512	1.262	0.6
8	There are enough people I feel close	0.375	0.484	0.512	1.263	0.3
9	I miss having people around me	0.260	0.439	1.092	2.193	0.2
10	I often feel rejected	0.140	0.347	2.074	5.301	0.4
11	I can call on my friends whenever I need them	0.265	0.441	1.064	2.133	0.2

**Table 4 healthcare-11-00489-t004:** Factor structure and factorial loads of the items in the two-factor model.

	Items	Initial Communalities	Factorial Loads	
			Factor 1 Emotional Loneliness	Factor 2 Social Loneliness
1	There is always someone I can talk to about my day-to-day problems	0.461		0.521
2	I miss having a really close friend	0.626	0.829	
3	I experience a general sense of emptiness	0.780	0.825	
4	There are plenty of people I can lean on when I have problems	0.704		0.760
5	I miss the pleasure of the company of others	0.776	1.001	
6	I find my circle of friends and acquaintances too limited	0.508	0.389	0.419
7	There are many people I can trust completely	0.646		0.778
8	There are enough people I feel close	0.761		0.971
9	I miss having people around me	0.737	0.928	
10	I often feel rejected	0.741	0.724	
11	I can call on my friends whenever I need them	0.657		0.864
	Cronbach’s alpha		0.776	0.892
	Cronbach’s Alpha of total scale		0.899	

Extraction method: Principal Component Analysis (PCA) with Robust Promin rotation. *n* = 957. Factorial loads higher than 0.30 are shown.

**Table 5 healthcare-11-00489-t005:** Comparison of the Confirmatory Factor Analysis models.

	Goodness of Fit Indexes				
	χ^2^	*p*-Value	CFI	TLI	RMSEA (CI 95%)
One-factor model	526.477	<0.001	0.875	0.844	0.111 (0.103–0.119)
Original two-factor model	247.250	<0.001	0.951	0.937	0.070 (0.062–0.079)
Two-factor model with item 6 loading to both factors	183.981	<0.001	0.966	0.955	0.059 (0.051–0.069)

CFI: Comparative Fit Index; TLI: Tucker–Lewis Index; RMSEA: Root-Mean-Square Error of Approximation; CI: Confidence Interval.

**Table 6 healthcare-11-00489-t006:** Discriminant validity of the DJGLS in the Mexican sample.

	Correlation		Discriminant Index	
	Social Loneliness	Emotional Loneliness	CR	AVE
Social Loneliness	1.000		0.986	0.923
Emotional Loneliness	0.319	1.000	1.012	1.017

CR: Composite Reliability; AVE: Average Variance Extracted.

**Table 7 healthcare-11-00489-t007:** Mean differences in Depressive Symptoms and Social Support between the Loneliness groups.

		Mean (*n*)	SD	t	df	*p*-Value
(a) Depressive Symptoms						
	No loneliness group	1.81 (906)	1.790	−21.500	1384.18	<0.001
	High/Very high loneliness group	4.55 (913)	3.415			
(b.1) Marital Support						
	No loneliness group	9.75 (459)	0.045	5.423	617.078	<0.001
	High/Very high loneliness group	9.27 (364)	0.074			
(b.2) Filial support						
	No loneliness group	9.81 (850)	0.030	10.941	1337.95	<0.001
	High/Very high loneliness group	9.15 (838)	0.052			
(b.3) Family support						
	No loneliness group	9.47 (705)	0.044	8.231	863.032	<0.001
	High/Very high loneliness group	8.76 (511)	0.074			

SD: standard deviation; df: degrees of freedom.

## Data Availability

All data used and/or analyzed during the current study are available from the corresponding author upon reasonable request (magudelo@unam.mx).

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
