# Peer review of "Psychometric Properties of the 11-Item De Jong Gierveld Loneliness Scale in a Representative Sample of Mexican Older Adults"

_healthcare, 2023, doi:10.3390/healthcare11040489_

Round 1

Reviewer 1 Report

Thank you for the well-written and very detailed article on the psychometric properties of the DJGLS.

Overall, I have only a few comments which will please be taken into account when revising the text.

Lines 47-48: Does the primary sources on UCLA and CELS get referenced in sources 11-15? If not, please still cite the primary sources.

Line 48: Please also write out the abbreviation DJGLS the first time in the long text.

Line 85: Which "abuse"?

From line 89: The collection method is missing here. Computer Assisted Personal Interview or did the interviewers use pen and paper to record the responses?

Were the participants informed in advance by mail/letter that an interviewer will be visiting them?

Line 108: To make it easier for international readers to understand, you should point out that the statements from the original scale were transformed into questions for the adaptation (presumably because they were asked by interviewers).

Line 168 vs. line 171: The sample size differs by 2 persons. What about the two missing ones?

Line 181: Please modify the table heading to "Descriptive statistics of the DJGLS items".

Table 4: A bracket is missing for the last confidence interval.

Line 242: groups

Line 302: Since it is also given as a limitation in the English literature of the DJGLS: Here the questions were asked by an interviewer. Accordingly, it must be discussed here whether the data collection method paper/pencil or CAPI (which is not entirely clear from the article so far) had or could have an influence on the data quality. In addition, social desirability always plays a role in face-to-face interviews. A social desirability bias is therefore quite conceivable.

Author Response

Thank you for the well-written and very detailed article on the psychometric properties of the DJGLS.

Overall, I have only a few comments which will please be taken into account when revising the text.

Thank you for taking the time to read our manuscript and your comments. In the following lines, a point by point answer to each of your observations is included.

Lines 47-48: Does the primary sources on UCLA and CELS get referenced in sources 11-15? If not, please still cite the primary sources.

The original UCLA Loneliness Scale (Version 3) was used, source 13 in the references is the original source of the DGLS.

Line 48: Please also write out the abbreviation DJGLS the first time in the long text.

This was corrected.

Line 85: Which "abuse"?

We now included the full name of the original project.

From line 89: The collection method is missing here. Computer Assisted Personal Interview or did the interviewers use pen and paper to record the responses?

This was corrected. In the methods section, we explain that the interviews were conducted face to face using a paper questionnaire.

Were the participants informed in advance by mail/letter that an interviewer will be visiting them?

No, the interviewers arrived the home of the older adults, presented the study and requested to carry out the interview at that moment. Homes were selected following the sample design.

Line 108: To make it easier for international readers to understand, you should point out that the statements from the original scale were transformed into questions for the adaptation (presumably because they were asked by interviewers).

This information was incorporated into materials and methods section.

Line 168 vs. line 171: The sample size differs by 2 persons. What about the two missing ones?

Suitable changes were made.

Line 181: Please modify the table heading to "Descriptive statistics of the DJGLS items".

Suitable changes were made.

Table 4: A bracket is missing for the last confidence interval.

Suitable changes were made.

Line 242: groups

Suitable changes were made.

Line 302: Since it is also given as a limitation in the English literature of the DJGLS: Here the questions were asked by an interviewer. Accordingly, it must be discussed here whether the data collection method paper/pencil or CAPI (which is not entirely clear from the article so far) had or could have an influence on the data quality. In addition, social desirability always plays a role in face-to-face interviews. A social desirability bias is therefore quite conceivable.

Thank you for this comment, we now specify in the methods section that interviews were conducted face to face using a paper questionnaire. Regarding social desirability, although it is an interesting point, through our study it is not possible to assess this bias. We have included a paragraph in the limitations section, addressing this as a potential limitation.

Reviewer 2 Report

I attach the suggestions for improvement for this paper.

Author Response

This is an article focused on an area with good justification and the need to deepen the subject "Psychometric properties of the 11-item De Jong Gierveld Loneliness scale in a representative sample of older Mexican adults". In general, most of the methods have been used to a good standard and have been well described. I have some comments and suggestions to help improve clarity in parts of the document:

Thank you for the time invested in reading our article. In the following lines, point by point changes in the document addressing your comments are included.

In the Summary: The summary is very brief and should include more relevant information:

- It would be interesting to include the selection criteria and what type of evidence was obtained.

- It would be interesting to include the time period for data collection

This was modified incorporating the most relevant information and the selection criteria of the study participants in the methods section

In the introduction:

- The introduction section should go into more relevant information, such as the items that make up the DJGLS questionnaire.

Information from the scale was incorporated.

- Lines 79-81. It would be interesting to include a clearly defined objective of the study... The objective of the study is... to help a better reading and comprehension...

The correction was made, both in the abstract and in the introduction.

In the Materials and Methods:

- This section should be improved and structured in depth with subsections to facilitate reading and understanding (study design, ethical considerations, setting and participants, instruments, procedure, statistical analysis). In addition, it would be interesting to include a flow diagram of the participants and specify the inclusion and exclusion criteria.

Thank you for your comment. The correction was made. Ethical considerations are found at the end of the document, as requested by the journal.

In the conclusion:

- The conclusions section is too short, it would be interesting to also include the strengths and weaknesses of this study.

Thank you for this, we included the limitations in section 4.1 of the discussion and we also expanded the conclusion.

Author Response

Thank you for your comments. We appreciate the time taken to read and comment our work. The following lines explain point by point changes made to the document.

  1. The data collection procedures should be developed to answer/address the following issues: administration through interview or self-administration; time of response; feedback about the comprehensibility of the items; existence of an informant consent; ethical board authorization.

Changes were addressed. Ethical considerations are found at the end of the document, as the journal requests it in the assigned format.

  1. I would suggest a clear description of DJGLS scale’s dichotomization.

A clearer description of the operationalization of the data is now provided.

  1. I found the use of Shapiro-Wilk test redundant since data is categorical/dichotomous.

The Shapiro-Wilk test was eliminated.

  1. The use of particular methods when running EFA and CFA with categorical/dichotomous data should be better justified, adding relevant and valid references which support those options.

References that adequately justify the different analyses were incorporated.

  1. The authors refer that construct, discriminant and convergent validity were tested. In literature, we can find different definitions of “validity” of a measure and also of the various types of validity (face, content, criterion, construct). Thus, the authors should indicate the respective references to justify the options made.

In psychometrics, the two essential metric characteristics to assess the precision of an instrument are reliability and validity. Reliability can be estimated by different means, including internal consistency. While validity is a key element both for the design of a questionnaire and to verify its usefulness. Content validity, criteria validity and construct validity can be estimated in different ways. Each of them provides evidence for the global validation of the instrument. In the present work, the assessment of content validity was not carried out because the DJGLS scale is an instrument that has been previously validated and is widely used. Criterion validity is the degree of correlation between an instrument and a gold standard. However, in our study, no other instrument to measure loneliness was applied to the population, thus comparisons to a gold standard measure were not possible.

  1. How did the authors replace the missing-values? When they present the items with higher % of missing-values the value presented after the respective item indicates the response rate (and not the % of missing-values).

Thank you for this, we now explain in the document that missing values ​​were not imputed. Likewise, the reference to rates by percentages was corrected.

  1. Fit comparison between the original two-model and the two-factor model with item 6 loading both factors is not presented in the CFA´s results.

Data are shown in table 5.